# New failure mechanism for evaluating ultimate inclined load adjacent to slope

**Hongwei Fang[1], Ning Wang[1], Yixiang Xu[2]\***

**1** School of Geomatics and Prospecting Engineering, Jilin Jianzhu University, Changchun, 130118, China,
**2** UNNC-NFTZ Blockchain Laboratory, School of Aerospace, University of Nottingham Ningbo, Ningbo, 315100, China

\* yixiang.xu@nottingham.edu.cn

**Data Availability Statement:** All relevant data are within the manuscript and its Supporting Information files.

**Funding:** The research was supported by the science and technology research project of the

## Abstract

A new failure mechanism is proposed for calculating the ultimate inclined load adjacent to the slope, i.e., the slope is in the limit state when the critical slope contour and the slope surface are at the critical position where two intersections will occur. The conventional view is that the critical slope contour calculated by the method of characteristics has only a concave shape. This study found that the critical slope contour changes from concave to convex when the inclined load imposed on the slope top surface increases. The feasibility of the proposed method is verified by the finite element limit analysis (FELA) and the definition of the ultimate load. The parametric analysis showed that the current method of characteristics (CMOC) overestimated the ultimate inclined load and gave an incorrect conclusion since it assumed larger failure models at a low strength ratio or large friction angle. The proposed method does not require assumption or search of the failure models, and it can solve the shortcomings of CMOC.

## 1 Introduction

Many structures, e.g., buildings, bridge abutments, and transmission line towers, are built near slopes, and the foundations of the structures are usually subjected to the inclined load. It is a complex problem to calculate the ultimate inclined load of a shallow foundation adjacent to the slope. The inclined load and slope reduce the bearing capacity of the soil [1]. Current research gives exact solutions or empirical equations for the effect of inclined loads on the bearing capacity of foundations on the horizontal ground to calculate the damage loads. However, there is no complete solution for inclined loads for foundations on slopes [2].

The foundation bearing capacity and slope stability are both related to the limit state of the system and should be equivalent in terms of failure mechanisms [3]. Thus, the determination of the failure mechanism and the limit state is a challenging problem in the study of the ultimate inclined load of the foundation adjacent to the slope. The logarithmic spiral failure model was assumed using the limit equilibrium method [4]. The failure models based on the incremental displacement vectors were searched using FELA, e.g., the influence of inclined and eccentric loading on the bearing capacity of a strip footing placed on the reinforced cohesionless soil slope by using lower bound FELA [5]; the finite element program OptumG2 was used to study the undrained bearing capacity of an inclined loaded strip footing near a

education department of Jilin province. (No. JJKH20220280KJ).

**Competing interests:** NO authors have competing interests

cohesive slope with a spatial variability of the undrained shear strength [6]. Fast Lagrangian analysis of the finite difference code continuum [7] was used to numerically analyze the bearing capacity of a strip footing near a cohesionless slope under a central inclined load. The determination of the failure model can be regarded as a nonlinear and nonsmooth global optimization problem. It is difficult to optimize the load problem with the existence of multiple local minima [8].

CMOC calculated the ultimate load starting from the stress state of the slope surface and assumed the outermost slip line as the critical slip surface [9–11]. According to the Mohr-Coulomb failure criterion, every slip line may be a slip surface. Only slip lines with a minimum safety factor are critical slip surfaces. The strength reduction method can directly obtain the failure modes, but the instability criteria need further study [12,13], e.g., there is no guidance on the selection of the convergence criteria or the optimal number of iterations, and the sharp point immediately is difficult to find when the displacement curve is relatively smooth. CMOC considered that the critical slope contour is concave [14–16], e.g., an instability criterion was proposed by [17] which is only applicable to the state where the critical slope contour intersects the slope toe. In this study, a convex critical slope contour was found and a failure mechanism was introduced to calculate the ultimate inclined load of the foundation adjacent to the slope. The influence of geometrical and mechanical parameters on the proposed method and the disadvantages of CMOC are studied.

## 2 Algorithm

### 2.1 Slip line equations

The equations of the slip line obtained by [14] are briefly introduced in this section. The relationship between stress components and principal stresses is as follows:

$$\sigma_x = \frac{1}{2}(\sigma_1 + \sigma_3) + \frac{1}{2}(\sigma_1 - \sigma_3)\cos 2\theta \tag{1}$$

$$\sigma_y = \frac{1}{2}(\sigma_1 + \sigma_3) - \frac{1}{2}(\sigma_1 - \sigma_3)\cos 2\theta \tag{2}$$

$$\tau_{xy} = \frac{1}{2}(\sigma_1 - \sigma_3)\sin 2\theta \tag{3}$$

where $\sigma_x$, $\sigma_y$, $\tau_{xy}$ and $\tau_{yx}$ represent the normal and shear stress in x and y directions, $\sigma_1$ and $\sigma_3$ are the maximum and minimum principal stresses, $\theta$ is the angle between $\sigma_1$ and the x-axis.

The formula of characteristic stress $\sigma$ is introduced using Mohr-Coulomb criterion:

$$\sigma = \frac{\sigma_1 + \sigma_3}{2} + c\cot\varphi \tag{4}$$

$$\sigma = \frac{\sigma_1 - \sigma_3}{2\sin\varphi} \tag{5}$$

where $c$ and $\varphi$ are cohesion and internal friction angle.

Substituting Eqs (4) and (5) into Eqs (1)–(3), and the expressions of the normal and shear stress are given as follows:

$$\sigma_x = \sigma(1 + \sin\varphi\cos 2\theta) - c\cot\varphi \tag{6}$$

$$\sigma_y = \sigma(1 - \sin\varphi\cos 2\theta) - c\cot\varphi \tag{7}$$

$$\tau_{xy} = \sigma\sin\varphi\sin2\theta \tag{8}$$

The seismic differential equations are given as follows:

$$\frac{\partial\sigma_x}{\partial x} + \frac{\partial\tau_{xy}}{\partial y} = 0 \tag{9}$$

$$\frac{\partial\tau_{yx}}{\partial x} + \frac{\partial\sigma_y}{\partial y} = \gamma \tag{10}$$

where $\gamma$ represents the unit weight.

The limit equilibrium equations can be obtained by substituting Eqs (6)–(8) into Eqs (9) and (10):

$$(1 + \sin\varphi\cos2\theta)\frac{\partial\sigma}{\partial x} + \sin\varphi\sin2\theta\frac{\partial\sigma}{\partial y} - 2\sigma\sin\varphi(\sin2\theta\frac{\partial\theta}{\partial x} - \cos2\theta\frac{\partial\theta}{\partial y}) = 0 \tag{11}$$

$$\sin\varphi\sin2\theta\frac{\partial\sigma}{\partial x} + (1 - \sin\varphi\cos2\theta)\frac{\partial\sigma}{\partial y} + 2\sigma\sin\varphi(\cos2\theta\frac{\partial\theta}{\partial x} + \sin2\theta\frac{\partial\theta}{\partial y}) = \gamma \tag{12}$$

Supplementary full differential equations:

$$d\sigma = \frac{\partial\sigma}{\partial x}dx + \frac{\partial\sigma}{\partial y}dy \tag{13}$$

$$d\theta = \frac{\partial\theta}{\partial x}dx + \frac{\partial\theta}{\partial y}dy \tag{14}$$

According to the definition of the characteristic lines, $\alpha$ and $\beta$ families of the characteristic line equations can be obtained by solving the Eqs (11)–(14):

$$\frac{dy}{dx} = \tan(\theta - \mu) \tag{15}$$

$$d\sigma - 2\sigma\tan\varphi d\theta = \gamma(dy - \tan\varphi dx) \tag{16}$$

$$\frac{dy}{dx} = \tan(\theta + \mu) \tag{17}$$

$$d\sigma + 2\sigma\tan\varphi d\theta = \gamma(dy + \tan\varphi dx) \tag{18}$$

where $\mu = \frac{\pi}{4} - \frac{\varphi}{2}$ is the average angle between two families of slip line.

The characteristic line Eqs (15)–(18) are approximately solved by the finite difference method:

$$\frac{y - y_\alpha}{x - x_\alpha} = \tan(\theta_\alpha - \mu) \tag{19}$$

$$(\sigma - \sigma_\alpha) - 2\sigma_\alpha(\theta - \theta_\alpha)\tan\varphi = \gamma[(y - y_\alpha) - (x - x_\alpha)\tan\varphi] \tag{20}$$

$$\frac{y - y_\beta}{x - x_\beta} = \tan(\theta_\beta + \mu) \tag{21}$$

$$(\sigma - \sigma_\beta) + 2\sigma_\beta(\theta - \theta_\beta)\tan\varphi = \gamma[(y - y_\beta) + (x - x_\beta)\tan\varphi] \tag{22}$$

The following Eqs (23)–(26) can be derived from Eqs (19)–(22). Note that the two equations in Eqs (24) or (26) yield the same result.

$$x = \frac{x_\alpha\tan(\theta_\alpha - \mu) - x_\beta\tan(\theta_\beta + \mu) - (y_\alpha - y_\beta)}{\tan(\theta_\alpha - \mu) - \tan(\theta_\beta + \mu)} \tag{23}$$

$$\begin{cases} y = (x - x_\alpha)\tan(\theta_\alpha - \mu) + y_\alpha \\ y = (x - x_\beta)\tan(\theta_\beta + \mu) + y_\beta \end{cases} \tag{24}$$

$$\theta = \frac{(\sigma_\beta - \sigma_\alpha) + 2(\sigma_\beta\theta_\beta + \sigma_\alpha\theta_\alpha)\tan\varphi + \gamma[(y_\alpha - y_\beta) + (2x - x_\alpha - x_\beta)\tan\varphi]}{2(\sigma_\beta + \sigma_\alpha)\tan\varphi} \tag{25}$$

$$\begin{cases} \sigma = \sigma_\alpha + 2\sigma_\alpha(\theta - \theta_\alpha)\tan\varphi + \gamma[(y - y_\alpha) - (x - x_\alpha)\tan\varphi] \\ \sigma = \sigma_\beta - 2\sigma_\beta(\theta - \theta_\beta)\tan\varphi + \gamma[(y - y_\beta) + (x - x_\beta)\tan\varphi] \end{cases} \tag{26}$$

According to the Eqs (23)–(26), the unknown point $M$ $(x, y, \theta, \sigma)$ in the slip line can be obtained by the known points $M_\alpha$ $(x_\alpha, y_\alpha, \theta_\alpha, \sigma_\alpha)$ and $M_\beta$ $(x_\beta, y_\beta, \theta_\beta, \sigma_\beta)$ in the $\alpha$ and $\beta$ families, and $(x, y)$ denotes the coordinate. The parameters $M_{ij}(x_{ij}, y_{ij}, \theta_{ij}, \sigma_{ij})$ in the critical slope contour (i.e., the slope surface at limit state) are follows (Fang et al., 2020):

$$x_{ij} = \frac{x_b\tan\theta_b - x'_\beta\tan(\theta'_\beta + \mu) + (y'_\beta - y_b)}{\tan\theta_b - \tan(\theta'_\beta + \mu)} \tag{27}$$

$$y_{ij} = (x_{ij} - x_b)\tan\theta_b + y_b \tag{28}$$

$$\theta_{ij} = \frac{(\sigma'_\beta - \sigma_b) + 2(\sigma'_\beta\theta'_\beta + \sigma_b\theta_b)\tan\varphi + \gamma[(y_b - y'_\beta) + (2x_{ij} - x_b - x'_\beta)\tan\varphi]}{2(\sigma'_\beta + \sigma_b) \cdot \tan\varphi} \tag{29}$$

$$\sigma_{ij} = \frac{c\cot\varphi}{1 - \sin\varphi} \tag{30}$$

where $M_b(x_b, y_b, \theta_b, \sigma_b)$ and $M'_\beta(x'_\beta, y'_\beta, \theta'_\beta, \sigma'_\beta)$ are the known points of the critical slope contour and $\beta$ family slip line.

## 2.2 Boundary condition of inclined load

The closer the inclined load is to the slope crest, the more unstable the slope is. Thus, this paper only studies the case where the distance between the inclined load and the slope crest is assumed to be zero. As shown in Fig 1, three kinds of boundary value problems, e.g., Cauchy boundary (Active zone OAB), Degenerative Riemann boundary (Transition zone OBC), and Mixed boundary (Passive zone OCD), are needed to solve for calculation of the slip line field and the critical slope contour (i.e., line OD).

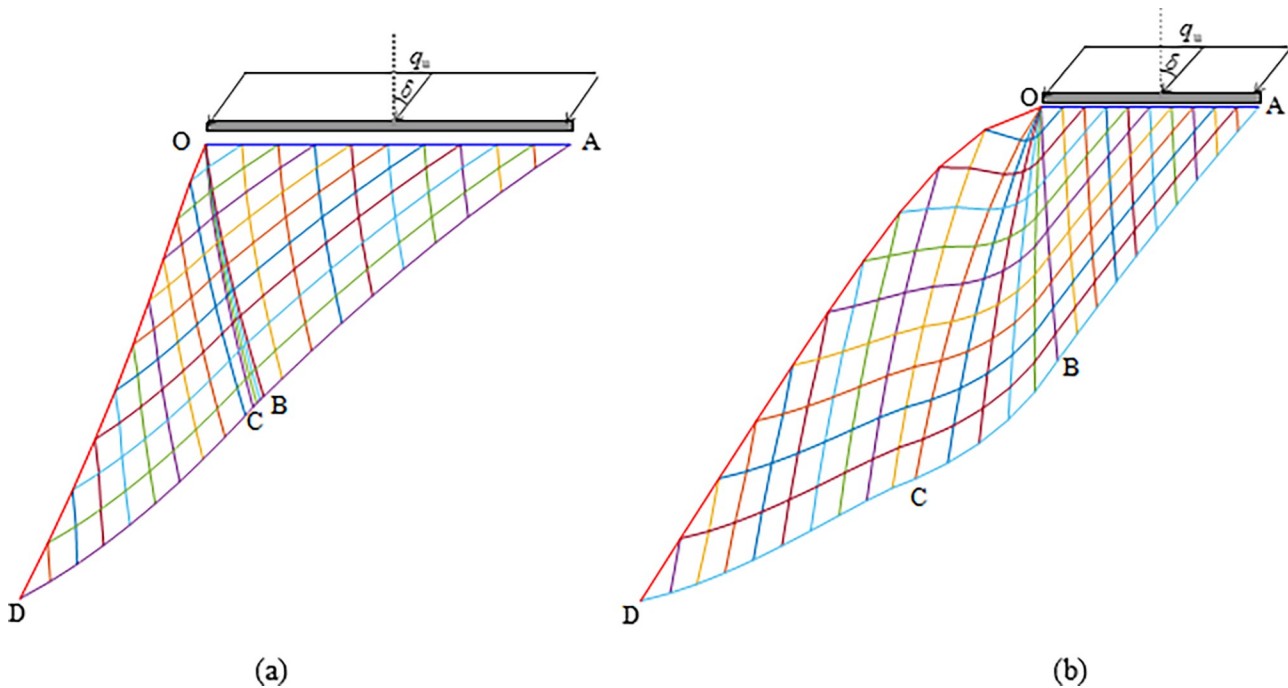

**Fig 1.** Slip line solution: (a) concave; (b) convex.

**2.2.1 Cauchy boundary.** As shown in Fig 2, $\theta_1$ and $\sigma_1$ of the $M_\alpha$ and $M_\beta$ points in the line OA can be derived using the Mohr-Coulomb failure criterion

$E_1G = EE_1\sin\delta = E_1F\sin\Delta_1 = EE_1\sin\varphi\sin\Delta_1$, i.e., $\sin\delta = \sin\varphi\sin\Delta_1$, and $\Delta_1 = \arcsin\left(\frac{\sin\delta}{\sin\varphi}\right)$.

Thus, the expression of $\theta_1$ is:

$$\theta_1 = \frac{1}{2}(\pi + 2\psi) = \frac{1}{2}(\pi + \delta + \Delta_1) \tag{31}$$

$\sigma_0 + c\cot\varphi = \sigma_1(1 + \sin\varphi\cos2\psi) = \sigma_1(1 + \sin\varphi\cos(\delta + \Delta_1))$, and $\sigma_0 + c\cot\varphi = q \cdot \cos\delta$, i.e., $q\cos\delta = \sigma_1(1 + \sin\varphi\cos(\delta + \Delta_1))$, where $\delta$ is the inclination angle, $q$ is the inclined load

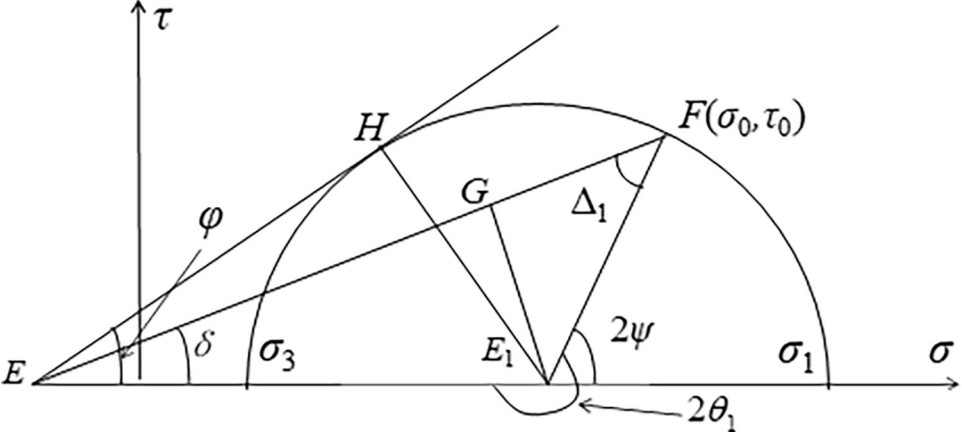

**Fig 2. Mohr circle of Cauchy boundary.**

imposed at the slope top surface. Thus, the expression of $\sigma_1$ is:

$$\sigma_1 = \frac{q\cos\delta}{1 + \sin\varphi\cos(\delta + \Delta_1)} \tag{32}$$

**2.2.2 Degenerative Riemann boundary.** The introduction of degenerative Riemann value is shown in **S1 Appendix**. The known $\sigma_2$ and $\theta_2$ of point O in zone OBC are:

$$\sigma_2 = \sigma_i = \sigma_2 e^{2(\theta_1 - \theta_i)\tan\varphi} \tag{33}$$

$$\theta_2 = \theta_i = \theta_1 + k\frac{\Delta\theta}{n} \tag{34}$$

$$\Delta\theta = \theta_3 - \theta_1 \tag{35}$$

where $\theta_3$ can be calculated by the Eq (36) in **Section 2.2.3**, $k = 0\sim n$, $n$ is the point partition of the Riemann boundary.

**2.2.3 Mixed boundary.** The first known point $M_b$ of the critical slope contour is point O in the zone OCD. According to Eq (30), $\sigma_b = \sigma_3 = \frac{c\cdot\cot\varphi}{1-\sin\varphi}$. $\theta_b = \theta_3$ can be obtained by substituting (30) into Eq (33):

$$\theta_b = \theta_3 = \theta_1 + \frac{1}{2}\cot\varphi \cdot \ln\frac{\sigma_1}{\sigma_3} \tag{36}$$

According to Eqs (35) and (36):

$$\Delta\theta = \frac{1}{2}\cot\varphi \cdot \ln\frac{\sigma_1}{\sigma_3} \tag{37}$$

According to Eq (30), $\sigma_3$ is a constant. According to Eq (32), $\sigma_1$ increases with $q$ increasing. Thus, $\Delta\theta$ increases as $\sigma_1$ and $q$ increase according to Eq (37). As shown in Fig 1(A) and 1(B), the critical slope contour OD changes from concave to convex with $\Delta\theta$ increasing.

# 3 Failure mechanism

## 3.1 Definition

For the convenience of calculation, the slope toe is defined as the coordinate origin. The position of the critical slope contour calculated from the method of characteristics varies with the increase of the inclined load $q_i = q_0 + i\cdot\Delta q$, where $q_0$ is the initial inclined load, $\Delta q$ is the inclined load increment, $i = 1, 2 \cdots\cdots n$. As shown in Fig 3(A), the failure mechanism is proposed for calculating the ultimate inclined load $q_u$: (1) the critical slope contour and the slope surface intersect at the slope crest (i.e., the first intersection) when $q_i < q_u$ (stable state); (2) the critical slope contour and the slope surface intersect at the slope surface (i.e., the second intersection) when $q_i > q_u$ (unstable state); (3) the critical slope contour is the critical boundary formed by transition point from one intersection to two intersections, i.e., $q_i$ is $q_u$ when the critical slope contour and the slope surface is at the critical position where two intersections will occur. The right-most slip line, i.e., the curve ABCD in Fig 1, is not the critical slip surface in this study. The instability criterion proposed by [17] is a special case of the proposed mechanism when the second intersection is the slope toe. The calculation flow chart is shown in Fig 3(B).

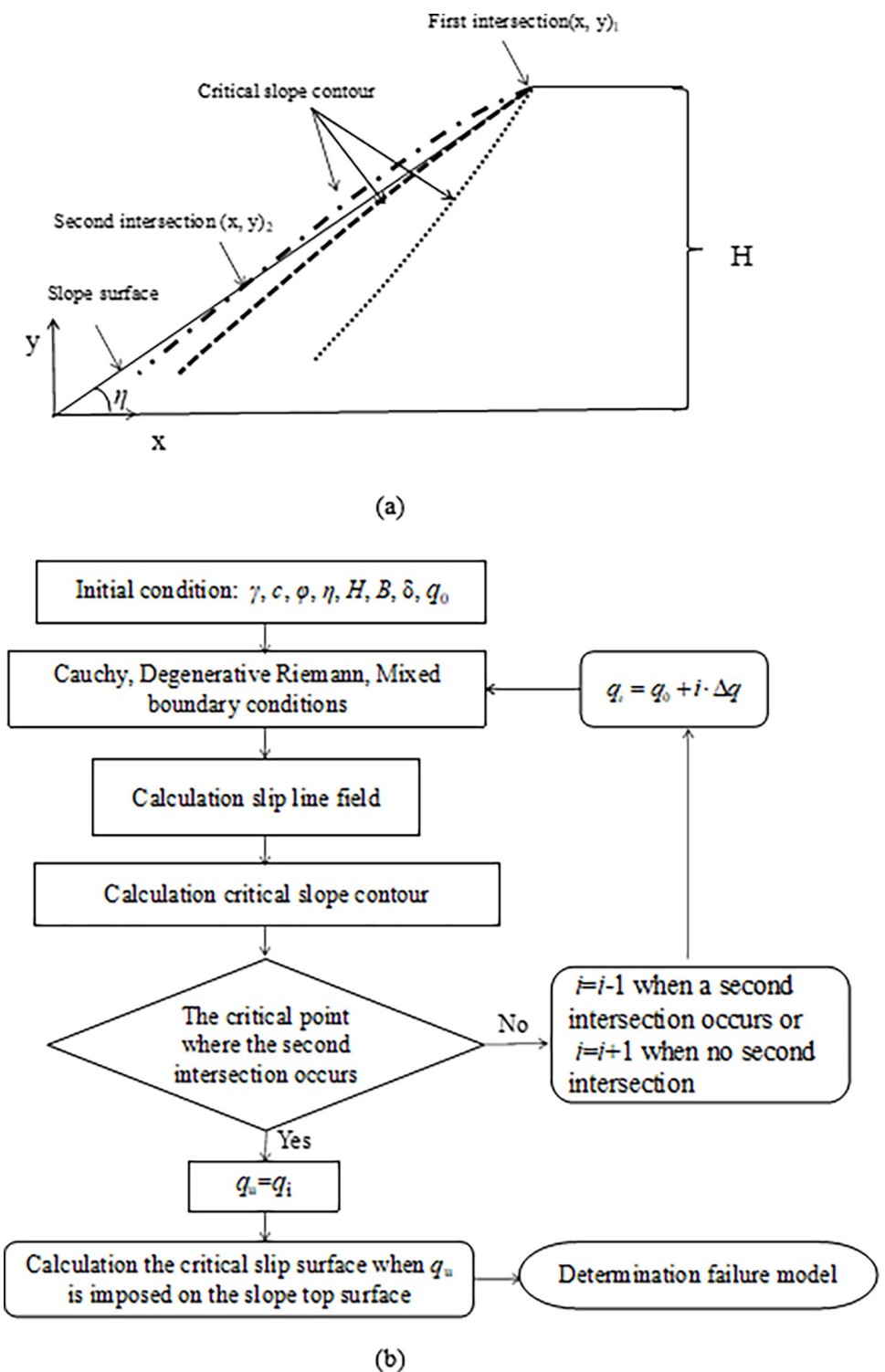

**Fig 3.** The proposed method: (a) the failure mechanism; (b) the calculation flow chart.

## 3.2 Verification

The parameters of the cases are $\gamma = 20kN/m^3$, $c = 20kPa$, $\varphi = 30^0$, $\eta = 30^0$, slope height H = 2m, B = 2m, $\delta = 15^0$, i.e., the strength ratio $c/\gamma B = 0.5$ and H/B = 1.0. The normalized ultimate inclined load factor is $N_u = q_u /\gamma B$. The slip line fields calculated by the proposed method are shown in Fig 4. As expected, the intersection of the critical slope contour and the slope surface changes from one to two with $N_i = q_i /\gamma B$ increasing. The critical slope contour and the slope surface have one intersection (i.e., the slope crest $(x, y)_1 = (3.46, 2.0)$) when $N_i$ increases from $N_1 = 2.575$ to $N_2 = 5.075$ (as shown in Fig 4(A) and 4(B)), and those have two intersections when $N_i > 5.075$, e.g., $(x, y)_2 = (1.51, 0.87)$ when $N_3 = 6.075$ (as shown in Fig 4(C)). According to Fig 3(A), $N_u = N_2 = 5.075$ (i.e., $q_u = 203kPa$). Fig 4(A)–4(C) show that $\Delta\theta$ increases with $N_i = q_i /\gamma B$ increasing, e.g., $\Delta\theta$ increases from $3.2^0$ to $45.79^0$ as $N_i$ increases from 2.575 to 6.075 (i.e., $q_i$ increases from 103kPa to 243 kPa). The critical slope contour changes from concave to convex as $\Delta\theta$ increases as shown in Fig 4(D).

In the following calculations, H = 5.0 (i.e., H/B = 2.5). To save space, the slip line field mesh is not given. Comparison between Figs 4(D) and 5(A), H/B varying does not affect the proposed failure mechanism, e.g., $N_u = 5.075$ ($q_u = 203kPa$) and the critical slope contour changes from concave to convex as $N_i$ increases from 2.575 to 6.075. $N_u$ values are 4.9, 4.6, and 5.1 calculated by CMOC and the low bound (LB) and the upper bound (UB)-FELA (Fig 8 in [10]). Thus, $N_u = 5.075$ calculated by the proposed method is consistent with the existing results and

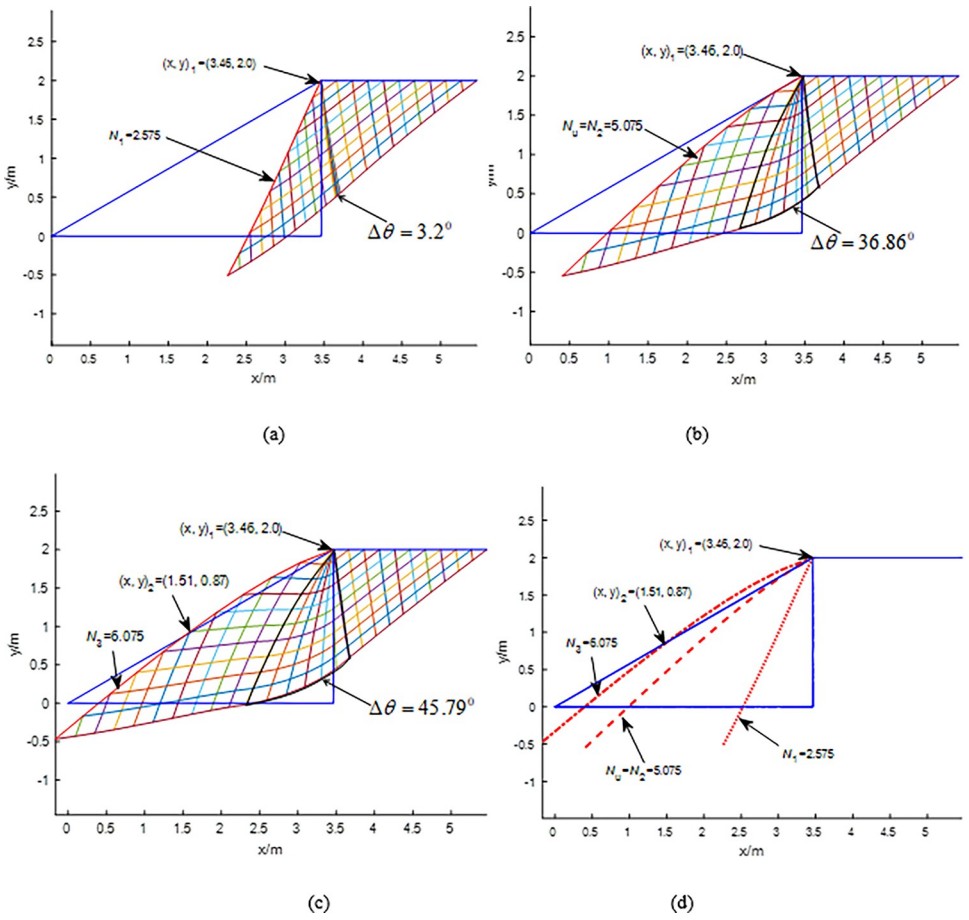

(a)

(b)

(c)

(d)

**Fig 4.** Calculation in the case: (a) $N_1 = 2.575$; (b) $N_u = N_2 = 5.075$; (c) $N_3 = 6.075$; (d) the proposed mechanism.

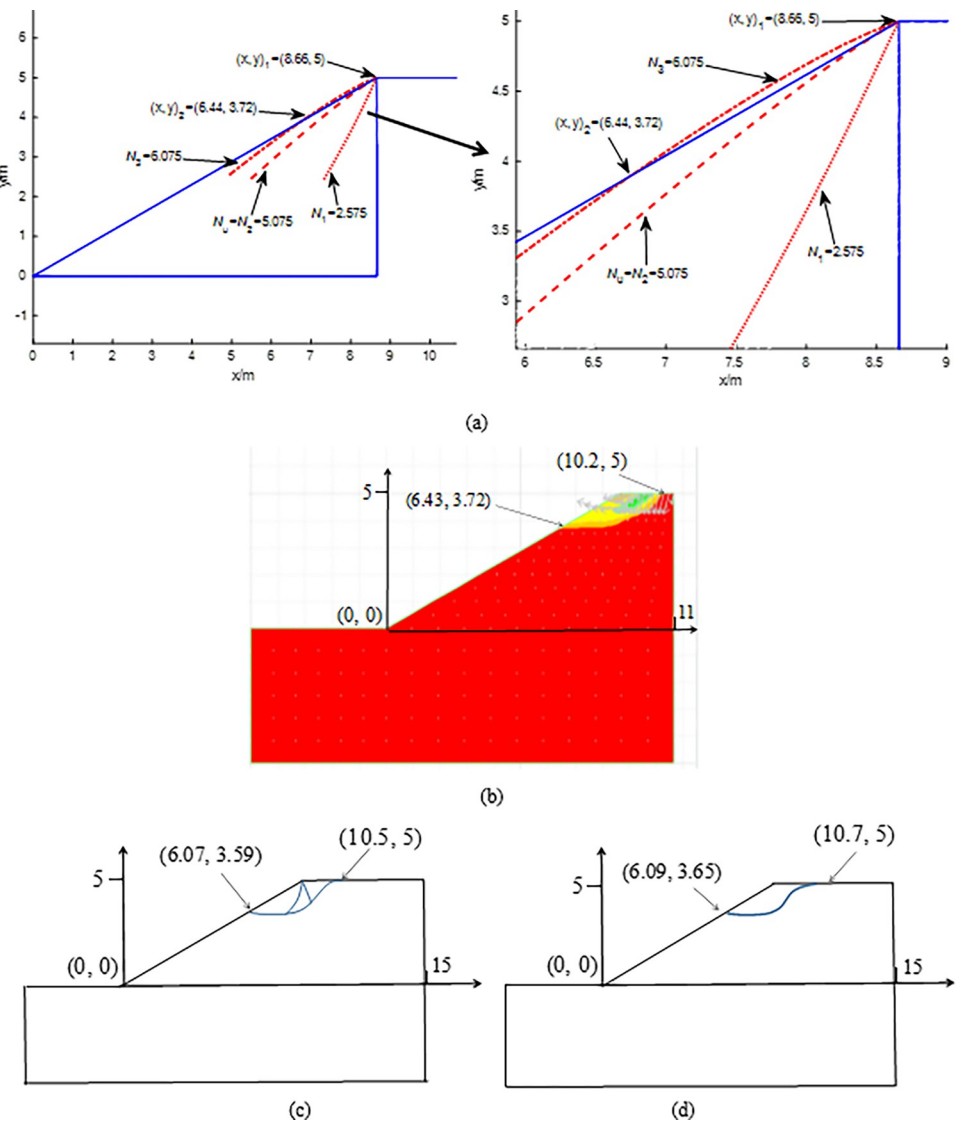

**Fig 5.** $N_u$ and the failure model: (a) the proposed failure mechanism; (b) the failure model of the proposed method; (c) the current slip line method (Fig 9(A) in [10]); (d) the LB-FELA (Fig 9(B) in [10]).

the errors are 3.4%, 9.3%, and 0.49%. When $N_u = 5.075$ ($q_u = 203$kPa) is imposed at the slope top surface, the failure model based on the shear strain rate is shown in Fig 5(B). Comparison between Fig 5(B)–5(D), the failure model of the proposed method is consistent with those of CMOC and LB-FELA.

# 4 Parameters study

## 4.1 φ variation

When H/B = 2.5, $c/\gamma B = 0.5$, φ varies from $20^0$ to $40^0$, $\eta = 30^0$, and $\delta$ varies from $0^0$ to $40^0$, Fig 6 (A) shows that $N_u$ values calculated by the proposed method decrease with $\delta$ increasing and φ decreasing. $N_u$ values obtained by the proposed method are consistent with those of UB or LB-FELA and CMOC (Fig 8 in [10]), except for the case of $\delta = 0^0$ and $\varphi = 40^0$.

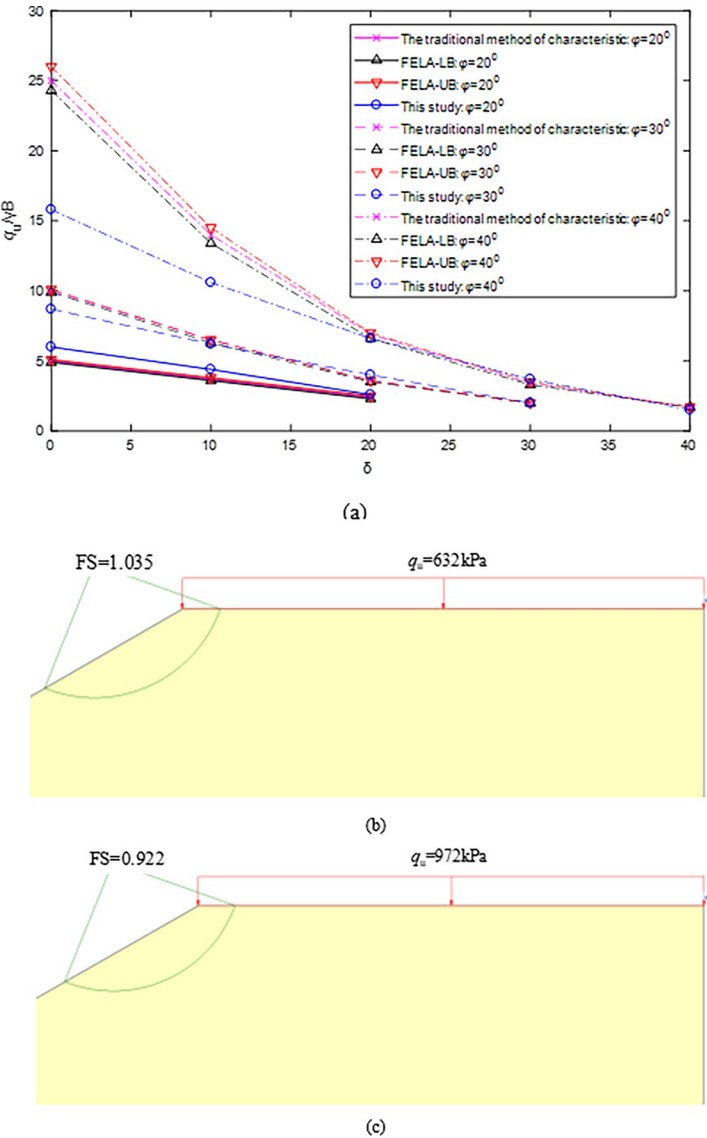

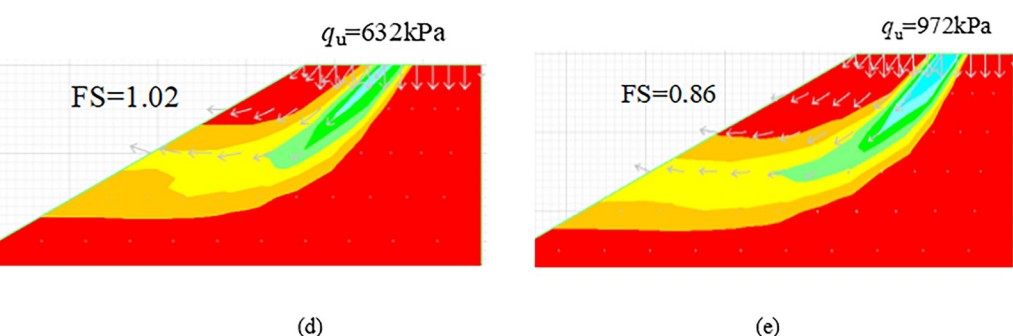

**Fig 6.** Influence of $\varphi$ on $N_u$, and calculated FS when $\varphi = 40^0$ and $\delta = 0^0$: (a) Comparison of $N_u$; (b) SLIED5.0 with $N_u = 15.8$; (c) SLIED5.0 with $N_u = 24.3$; (d) FLAC7.0 with $N_u = 15.8$; (e) FLAC7.0 with $N_u = 24.3$.

When $\delta = 0^0$ and $\varphi = 40^0$, $N_u$ values are 15.8 ($q_u$ = 632kPa) and 24.3 ($q_u$ = 972kPa) calculated by the proposed method and CMOC. As shown in Fig 6(B)–6(E), The safety factor (FSs) are calculated using SLIED5.0 and FLAC7.0 when $N_u$ = 15.8 ($q_u$ = 632kPa) and $N_u$ = 24.3 ($q_u$ = 972kPa) are imposed at the slope top surface. FS is equal to 1.0 when the ultimate inclined load is imposed on the slope top surface. Compared with CMOC (i.e., FS = 0.922 and 0.86), FS = 1.035 and 1.02 calculated with the proposed method are more closed to 1.0. The reason is that the failure mode becomes shallower with $\varphi$ increasing [18], e.g., $\varphi$ increases from $20^0$ to $40^0$ in this case. CMOC assumed that the outermost slip line is the critical slip surface to obtain larger failure modes and overestimated $q_u$.

## 4.2 η and c/γB variation

When H/B = 2.5, $c/\gamma B$ = 0.25, $\varphi = 30^0$, $\eta$ varies from $15^0$ to $60^0$, and tan($\delta$) varies from 0 to 0.5, comparison of $N_u$ values calculated by the proposed method and FELA are shown in Fig 7(A), where UB and LB-FELA solutions are calculated using OptumG2 [19]. For $\eta = 15^0$, $30^0$, and $60^0$, $N_u$ values calculated by the proposed method are almost between those of UB and LB-FELA, except in the cases of tan($\delta$) = 0.3 and $\eta = 60^0$. When tan($\delta$) = 0.3 and $\eta = 60^0$, $N_u$ = 1.1 and 1.06 are calculated by the proposed method and the UB-FELA, and the error is 3.6%. For $\eta = 45^0$, the result of the proposed method is slightly larger than those of the UB-FELA and the maximum error is 7.4%, e.g., $N_u$ = 3.2 and 3.0 are calculated by the proposed method, and the UB-FELA when tan($\delta$) is 0.

Fig 7(B) shows that $N_u$ calculated by CMOC (Fig 11(b) in [10]) are much larger than those of the proposed method. Thus, CMOC overestimated $N_u$, e.g., $N_u$ values obtained by CMOC are 3.4, 3.4, 3.4, and 2.1 for $\eta = 15^0$, 30, $45^0$, and $60^0$ when tan($\delta$) is 0.5. Under the same condition, $N_u$ values obtained by the proposed method are 1.7, 1.3, 0.9, and 0.7. The proposed method is more reasonable since the $N_u$ values of the proposed method are close to those of the FELA as shown in Fig 7(A). Table 1 shows that the FS related to the $N_u$ of the proposed method, calculated by SLIED5.0, is close to 1.0.

Fig 8 shows that the $N_u$ values obtained by the proposed method are close to CMOC and the mean of the UB and LB-FELA when H/B = 2.5, $c/\gamma B$ = 0.5, $\varphi = 30^0$, $\eta = 30^0$, and $\delta$ vary from $0^0$ to $30^0$. When $c/\gamma B$ = 0.25, the $N_u$ values of the proposed method are close to the mean of the UB and LB-FELA. The larger the strength of the slope is, the higher the ultimate inclined load is. $N_u$ increases with $c/\gamma B$ increasing. Thus, CMOC (i.e., Rigid-plastic face failure) gave an unreasonable result since $N_u$ decreases as $c/\gamma B$ increases from 0.25 to 0.5.

## 5 Discussion

This section discusses in detail the reason why the current method misjudges the ultimate inclination load when $c/\gamma B$ = 0.25 in Fig 8. As shown in Fig 8, $N_u$ values are 2.525 and 7.65 ($q_u$ = 101kPa and 306kPa) calculated by the proposed method and CMOC when H = 5m, $c/\gamma B$ = 0.25, $\varphi = 30^0$, $\eta = 30^0$, $\delta = 15^0$. Fig 9(A) shows that Face bearing capacity failure (see Fig 14(c) in Li et al., 2021 [10]) was assumed by CMOC. Fig 9(B)–9(E) show that the failure models were obtained using SLIDE5.0 and FLAC7.0 when $q_u$ = 101kPa, 306kPa are imposed at the slope top surface.

As shown in Fig 9(A), the position of the critical slip surface calculated by CMOC is the foundation width (i.e., B = 2m). When using SLIDE5.0, the slide-out point coordinates of the slope top and the slope surface corresponding to $q_u$ = 101kPa, 306kPa are (9.3, 5), (7.5, 4.3) and (9.1, 5), (7.4, 4.3), while using FLAC7.0, the corresponding coordinates are (9.98, 5.1), (7.1, 4.1) and (9.72, 5.1), (6.7, 3.9). FSs are 1.057 and 1.05 related to qu = 101kPa, which are close to 1.0. FSs are 0.677 and 0.66 related to qu = 306kPa. Thus, CMOC (i.e., Rigid-plastic

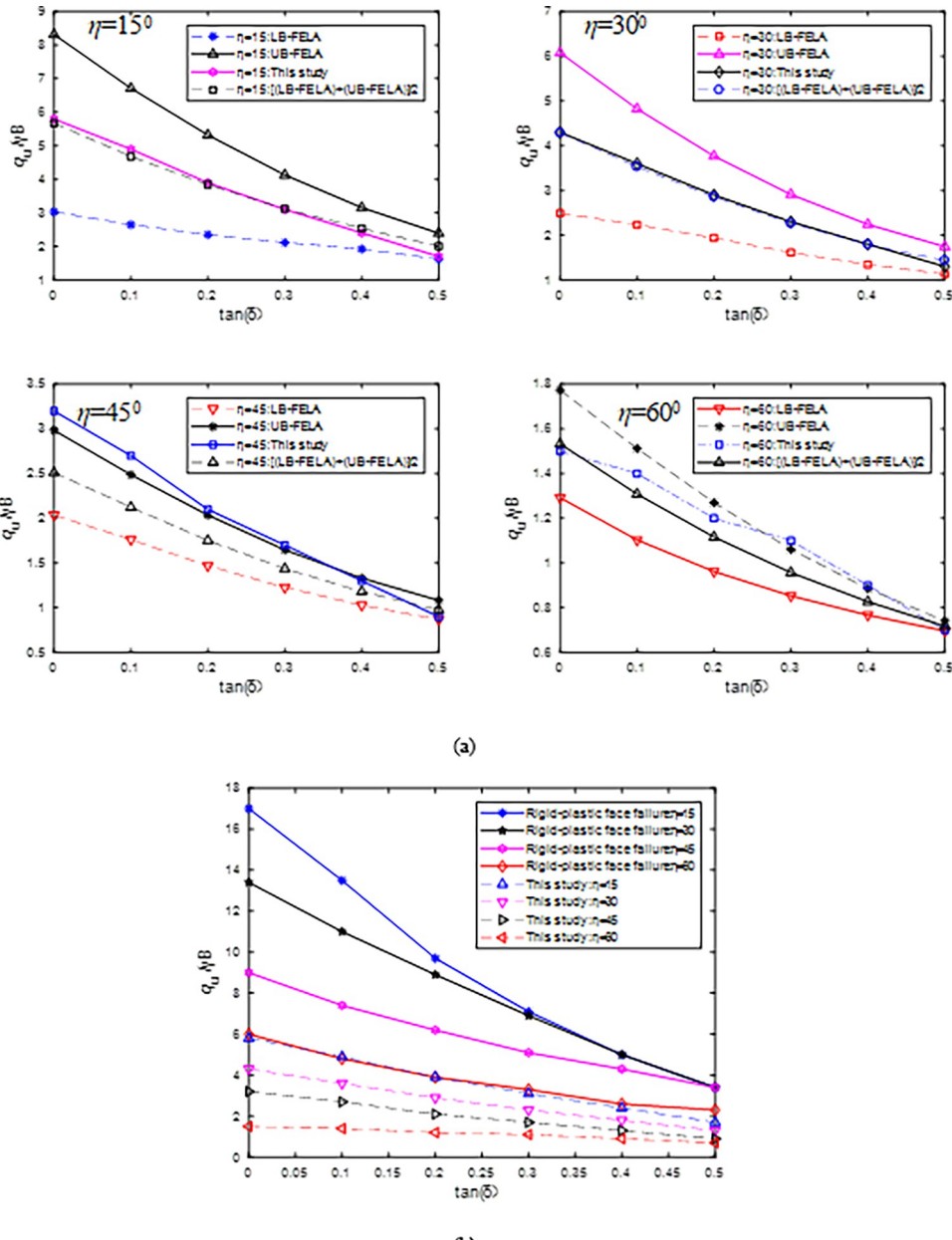

**Fig 7.** Influence of $\eta$ on $N_u$: (a) the proposed method and FELA; (b)the proposed method and CMOC.

face failure) obtains the large failure models and overestimates $q_u$ when the strength ratio becomes smaller. The reason is that the critical slip surface becomes shallower as the strength ratio decreases [18,20].

## 6 Conclusion

1. The boundary value problems were derived for calculating the slip lines and the critical slope contour when the inclined load is imposed at the slope top surface. The angle ($\Delta\theta$) between the maximum principal stress and the x-axis in the Degenerative Riemann

**Table 1. FS of the proposed method.**

| tan($\delta$) | | 0 | 0.1 | 0.2 | 0.3 | 0.4 | 0.5 |
|---|---|---|---|---|---|---|---|
| $\eta = 15^0$ | $N_u$ | 5.8 | 4.9 | 3.9 | 3.1 | 2.4 | 1.7 |
| | FS | 0.98 | 0.94 | 0.93 | 0.92 | 0.96 | 1.05 |
| $\eta = 30^0$ | $N_u$ | 4.3 | 3.6 | 2.9 | 2.3 | 1.8 | 1.3 |
| | FS | 1.02 | 1.03 | 1.04 | 1.04 | 1.08 | 0.99 |
| $\eta = 45^0$ | $N_u$ | 3.2 | 2.7 | 2.1 | 1.7 | 1.3 | 0.9 |
| | FS | 0.96 | 0.95 | 0.97 | 0.99 | 1.05 | 1.03 |
| $\eta = 60^0$ | $N_u$ | 1.5 | 1.4 | 1.2 | 1.1 | 0.9 | 0.7 |
| | FS | 0.91 | 0.90 | 0.90 | 0.90 | 0.91 | 0.96 |

boundary (Transition zone) increases as the inclined load ($q$) imposed on the slope top surface increases. The critical slope contour changes from concave to convex when $\Delta\theta$ and $q$ increase.

2. The critical slope contour shifts from the inside of the slope to the outside of the slope as $q$ increases. The critical slope contour and slope surface have one intersection when the critical slope contour is inside the slope (stable state), and they have two intersections when the critical slope contour is outside the slope (unstable state). The critical slope contour and the slope surface are at the critical position where two intersections will occur, and $q$ is the ultimate inclined load $q_u$.

3. When the strength ratio is large (e.g., $c/\gamma B = 0.5$), $q_u$ and the failure model calculated by the proposed method are consistent with those of CMOC and LB-FELA. According to the

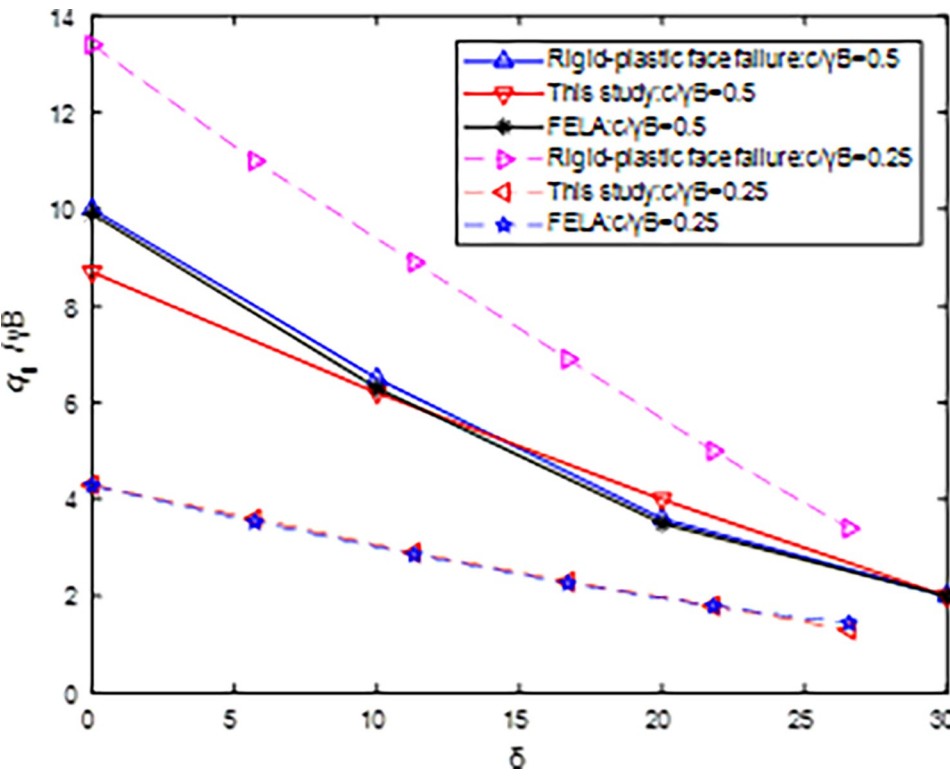

**Fig 8. Influence of $c/\gamma B$ on $N_u$ with H/B = 2.5, $\varphi = 30^0$, $\eta = 30^0$.**

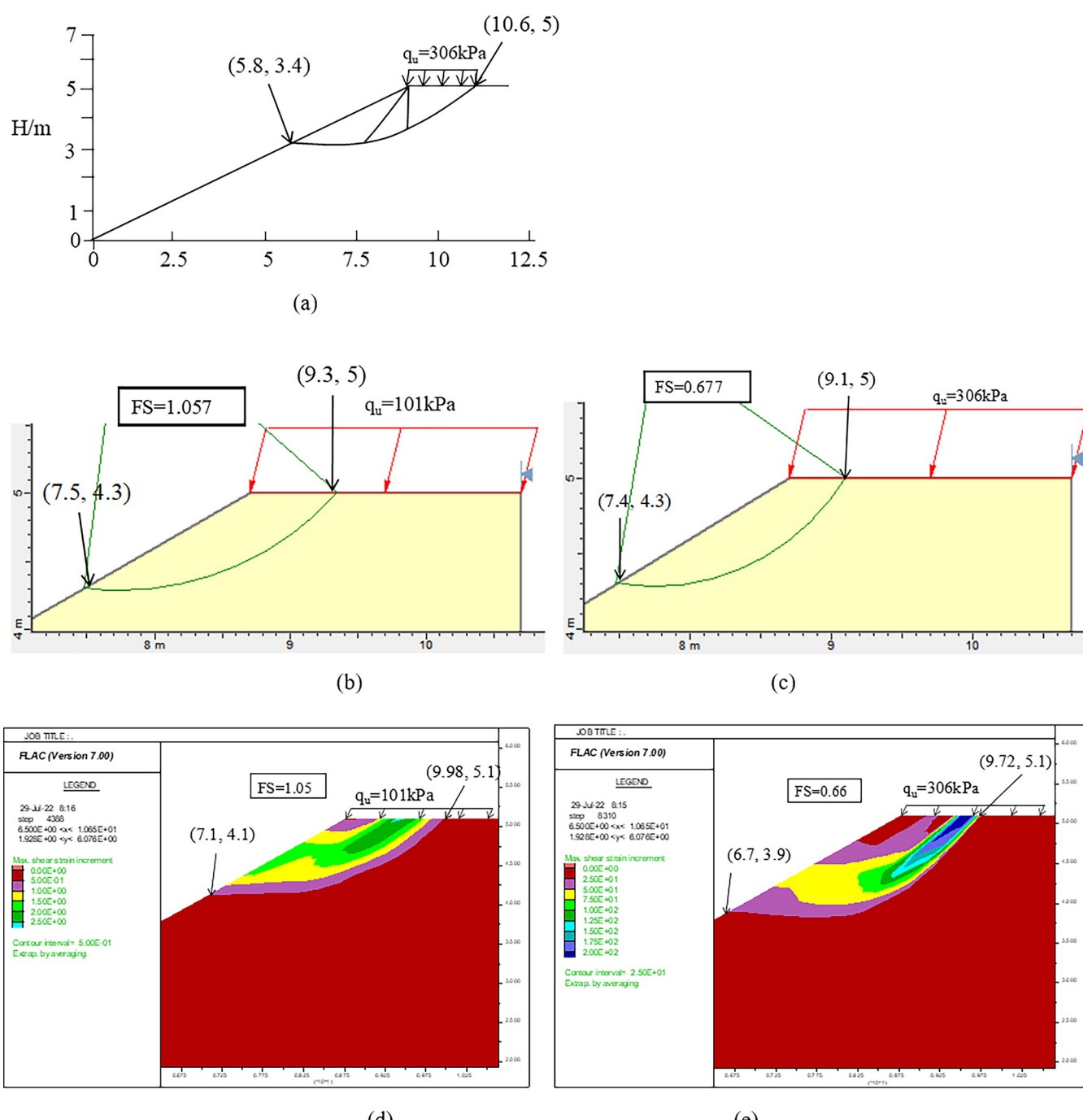

**Fig 9.** Critical slip surface: (a) Face bearing capacity failure; (b) Bishop by the proposed method; (c) Bishop by Face bearing capacity failure; (d) FLAC by the proposed method; (e) FLAC by Face bearing capacity failure.

definition of the ultimate inclined load (i.e., that the safety factor is equal to 1.0 when the ultimate inclined load is imposed on the slope top surface), the proposed method is more reasonable since the safety factor calculated by the proposed method is close to 1.0. The proposed method is close to the FELA when the strength ratio is small (e.g., $c/\gamma B = 0.25$).

4. CMOC overestimated $q_u$ or gave an unreasonable result when the friction angle is large (e.g., $\varphi = 40^0$) and the strength ratio is small (e.g., $c/\gamma B = 0.25$). The reason is that CMOC assumed the outermost slip line as the critical slip surface to obtain a larger failure model since the failure model becomes shallower with the friction angle ($\varphi$) increasing and the strength ratio decreasing.

5. CMOC considers that the critical slope contour is concave. This study finds that the critical slope contour is convex when the inclined load on the top of the slope increases. The proposed method does not need to assume or search the failure models, and it can solve the difficult problem that the failure models are not easy to determine. The proposed method gives the criterion of slope in the limit state without the strength parameters reduction, and it is simple and robust.

## Supporting information

**S1 Appendix.**
(DOCX)

## Author Contributions

**Conceptualization:** Hongwei Fang.

**Data curation:** Ning Wang.

**Methodology:** Hongwei Fang.

**Software:** Ning Wang.

**Writing – original draft:** Yixiang Xu.

**Writing – review & editing:** Yixiang Xu.

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
