## [Decision Letter · Decision Letter 0]

29 May 2023

PONE-D-23-08153New failure mechanism for evaluating ultimate inclined load adjacent to slopePLOS ONE

Dear Dr. Xu,

Thank you for submitting your manuscript to PLOS ONE. After careful consideration, we feel that it has merit but does not fully meet PLOS ONE’s publication criteria as it currently stands. Therefore, we invite you to submit a revised version of the manuscript that addresses the points raised during the review process.

The points arised by the Reviewer in respect to the accuracy of the results are curritial part of this study. Therefore I am encouredging the Authors to revise the study in respect to this perspective.

We look forward to receiving your revised manuscript.

Kind regards,

Abdullah Ekinci, PhD

Academic Editor

PLOS ONE

Journal Requirements:

“The research was supported by the science and technology research project of the education department of Jilin province. (No.JJKH20220280KJ).”

6. Please upload a copy of Figures 11 and 14, to which you refer in your text on page 13 and 15. If the figure is no longer to be included as part of the submission please remove all reference to it within the text.

Additional Editor Comments:

The points arise by the Reviewer in respect to the accuracy of the results are crucial part of this study. Therefore I am encouraging the Authors to revise the study in respect to this perspective.

Reviewers' comments:

Reviewer's Responses to Questions

**Comments to the Author**

1. Is the manuscript technically sound, and do the data support the conclusions?

Reviewer #1: No

Reviewer #2: Yes

2. Has the statistical analysis been performed appropriately and rigorously? 

Reviewer #1: No

Reviewer #2: Yes

3. Have the authors made all data underlying the findings in their manuscript fully available?

Reviewer #1: No

Reviewer #2: Yes

4. Is the manuscript presented in an intelligible fashion and written in standard English?

Reviewer #1: No

Reviewer #2: Yes

5. Review Comments to the Author

Reviewer #1: The paper proposes a new failure mechanism for calculating the ultimate inclined load adjacent to the slope. The topic is interesting, and the model could be useful for engineering practice. However, the paper lacks geotechnical discussion in general. Furthermore, the reasons for the selected range of geotechnical properties are not given.

Besides, the model proposes the evaluation of convex critical slip surfaces. However, the critical slip surfaces presented in Figs. 6 and 9 are concave. The manuscript and methods are interesting. However, the quality must be improved to be considered for publication. The mechanisms involved in slope stability and the choices made for modelling have to be better explained. Also, it is crucial to clearly state the limitations of the proposed model. Moreover, the numerical model is not compared to experimental data, so the accuracy of the results could not be assessed.

Reviewer #2: The paper is well organized and written in good language.

1-Please provide an additional explaation for the Riemann boundary, state the factorized conditions in thetab, the differecne between the Riemann and mixed boundary condition must be clearly defined, whcih factors must be considered in order to decrease the inlination or failure

2-please add the standard limitation for the FS value, comparebthe methods which shows lower FS and how the current proposed model could be classified as good over the other methods

3-add a brief recommendation section

6. PLOS authors have the option to publish the peer review history of their article (what does this mean?). If published, this will include your full peer review and any attached files.

Reviewer #1: No

Reviewer #2: **Yes: **Ertug Aydin

---

## [Author Response · Author response to Decision Letter 0]

5 Jun 2023

Author's Reply to the Review Report

We would like to thank the Reviewers for their comments and constructive suggestions which have greatly helped improve the paper. We have considered these comments and revised the manuscript accordingly. All changes in the revision are indicated in blue. 

Reviewer #1

(1) The paper proposes a new failure mechanism for calculating the ultimate inclined load adjacent to the slope. The topic is interesting, and the model could be useful for engineering practice. However, the paper lacks geotechnical discussion in general. Furthermore, the reasons for the selected range of geotechnical properties are not given.

Answer:

We have revised the paper such that more geotechnical discussion or applications are laid out. See new Figs. 6 (b) and 7(e). For calculation the ultimate inclined load adjacent to slope, the geotechnical properties include the unit weight (γ), cohesion (c) and internal friction angle (φ), footing width (B), slope height (H), slope angle (η), the inclination angle (δ). This paper has comprehensively considered the above geotechnical properties, e.g., H/B varying from 1 to 2.5 in lines 177, 191; c/γB varying from 0.5 to 0.25 in lines 177, 239, and section 4.3; φ varying from 200 to 400 , see line 209 in section 4.1; η varying from 150 to 600, see line 239 in section 4.2; δ varying from 00 to 400 in Figs. 6, 7 and 8. According to the results of this study with the above geotechnical properties, the selected range of geotechnical properties are very general and can be applicable to a wide range of engineering practice. 

(2) Besides, the model proposes the evaluation of convex critical slip surfaces. However, the critical slip surfaces presented in Figs. 6 and 9 are concave.

Answer:

Note that the critical slope contour (i.e., the line OD in Fig.1) is not the critical slip surface, see lines 165-167. Slip line field theory considered that the critical slope contour is concave [14-17]. The convex critical slope contour OD is found in this study. Section 2.2.3 and Figure 4 give the formation mechanism of the critical slope contour changing from a concave to a convex as the load becomes larger. The proposed method is to calculate the ultimate inclined load first and then determine the critical slip surface (i.e., the failure model). The calculation flow chart is shown in Figure 3(b). 

Figure 5 (a) gives the calculation diagram of the ultimate load as the critical slope contour changes from concave to convex. Figure 5 (b) gives the critical slip surface (concave) of the proposed method using the shear strain rate. Therefore, the critical slip surfaces in Figs. 6 and 9 are not the critical slope contour. Figure 5 (c), (d), Figure 6 (c), (d), (e), (f), and Figure 9 all indicate the critical slip surface are concave using SLIDE (Bishop method) and FLAC (Finite difference method). Therefore, the critical slip surface (concave) is not contradictory to the critical slope contour from concave to convex curve proposed in this paper.

(3)The manuscript and methods are interesting. However, the quality must be improved to be considered for publication. The mechanisms involved in slope stability and the choices made for modelling have to be better explained. 

Answer:

The legend of the expression of the proposed failure mechanism, i.e., three states (i.e., unstable, limiting equilibrium, stable states) corresponding to the different positions of the critical slope contour, is added in Figure 3(a). The expression statement of the proposed failure mechanism is modified in the manuscript, see line 165, 303-304. The selection judgment step in Fig. 3(b) is modified to make it clear that the load imposed at the slope top surface is the ultimate inclined load when the critical slope contour is tangent to the slope surface.

(4) Also, it is crucial to clearly state the limitations of the proposed model. Moreover, the numerical model is not compared to experimental data, so the accuracy of the results could not be assessed.

Answer:

The limitations of the proposed method is added in line 318, i.e., the proposed method needs further study for the non-homogeneous case. The feasibility of the proposed method is verified by the definition of the ultimate load, i.e., FSs calculated using SLIED5.0 (Bishop method) and FLAC7.0 (Finite difference method) are close to 1.0 when the ultimate inclined load is imposed at the slope top surface, see Figure 6 (c), (e), Figure 9 (b), (d), and Table 1. 

In the Revised Manuscript, the UB and LB-FELA solution calculated using OptumG2 (see line 213) are added in new Figures 6 (b), 7 (e). 

In new Figure 6 (b), Nu values calculated by the proposed method are close to those obtained from UB and LB-FELA using OptumG2. However, Nu values calculated by the current method of characteristics (CMOC) and FELA in [10] are close to those of the UB-FELA using OptumG2 when φ is 400. 

Fig.7(e) shows that Nu values calculated by CMOC are much larger than the values of UB and LB-FELA using OptumG2. However, Nu values calculated by the proposed method are close to the mean values of the UB and LB-FELA using OptumG2. 

The larger the strength of the slope is, the higher the ultimate inclined load is. Thus, Nu increases with c/γB increasing. Fig.8 shows that CMOC gave an unreasonable result because Nu decreases as c/γB increases from 0.25 to 0.5. Nu values calculated by the proposed method are close to the mean values of the UB and LB-FELA using OptumG2 with c/γB = 0.25, 0.5.

Comparing with the existing methods, e.g., SLIED5.0 (Bishop method), FLAC7.0 (Finite difference method) and the UB and LB-FELA using OptumG2, the proposed method can get the reasonable ultimate inclined load. The model experiments will be studied in the future.

Reviewer #2

(1) Please provide an additional explation for the Riemann boundary, state the factorized conditions in thetab, the differecne between the Riemann and mixed boundary condition must be clearly defined, whcih factors must be considered in order to decrease the inlination or failure

Answer:

Equation 33 has a small error, which was corrected, see line 140. To make the difference between the Riemann and mixed boundary condition clearer, a new equation 36 has been added, see line 148.

(2) please add the standard limitation for the FS value, compared the methods which shows lower FS and how the current proposed model could be classified as good over the other methods

Answer:

According to the definition of the ultimate load, i.e., FS is close to 1.0 when the ultimate inclined load is imposed at the slope top surface. As shown in Figure 6 (c), (e), Figure 9 (b), (d), and Table 1, FSs of the failure model calculated by the proposed method are close to 1.0. However, FSs calculated by CMOC are smaller than 1.0 as shown in Figure 6 (d), (f), and Figure 9 (c), (e). Thus, the ultimate inclined loads calculated by the proposed method are more reasonable, and it is better than CMOC.

(3) add a brief recommendation section

Answer:

The recommendation section is added in lines 319-323.

Journal Requirements:

Answer:

The format has been revised as required, see line 4.

“The research was supported by the science and technology research project of the education department of Jilin province. (No.JJKH20220280KJ).”

Answer:

The role of the fund is described in the text, see line 326.

Upon re-submitting your revised manuscript, please upload your study’s minimal underlying data set as either Supporting Information files or to a stable, public repository and include the relevant URLs, DOIs, or accession numbers within your revised cover letter. For a list of acceptable repositories, please see http://journals.plos.org/plosone/s/data-availability# loc-recommended-repositories. Any potentially identifying patient information must be fully anonymized.

Answer:

All the data used in the study are generated by using Matlab by inputing various variables. We have included a matlab file in the Revised manuscript and we believe all readers can replicate our results by using the program by inputing the relevant variables we have used in the manuscript. See Revised Cover letter.

Answer:

All the data used in the study are generated by using Matlab by inputing various variables. We have included a matlab file in the Revised manuscript and we believe all readers can replicate our results by using the program by inputing the relevant variables we have used in the manuscript. See lines 389-708 in the Revised manuscript. 

Answer:

 The right diagram is a enlarged diagram from the left diagram in Figure 5(a). The word “Enlarge” was added in Revised manuscript in Figure 5(a). Figure 7(a) is replaced by four figures, i.e., Figure 7(a), (b), (c), (d).

6. Please upload a copy of Figures 11 and 14, to which you refer in your text on page 13 and 15. If the figure is no longer to be included as part of the submission please remove all reference to it within the text.

Answer:

 The copies of Figures 11 and 14 in my text on page 13 and 15 are as follow. The reference is remove.

The copy of Figure 11(b) in [10], i.e., the values of CMOC in Figure 7(e) in Revised manuscript

The copy of Figure 14(c) in [10], i.e., Face bearing capacity failure in Figure 9(a) in Revised manuscript

---

## [Decision Letter · Decision Letter 1]

10 Jul 2023

New failure mechanism for evaluating ultimate inclined load adjacent to slope

PONE-D-23-08153R1

Dear Dr. Xu,

We’re pleased to inform you that your manuscript has been judged scientifically suitable for publication and will be formally accepted for publication once it meets all outstanding technical requirements.

Kind regards,

Ahmed Mancy Mosa, Ph.D.

Academic Editor

PLOS ONE

Additional Editor Comments (optional):

Reviewers' comments:

Reviewer's Responses to Questions

**Comments to the Author**

1. If the authors have adequately addressed your comments raised in a previous round of review and you feel that this manuscript is now acceptable for publication, you may indicate that here to bypass the “Comments to the Author” section, enter your conflict of interest statement in the “Confidential to Editor” section, and submit your "Accept" recommendation.

Reviewer #2: All comments have been addressed

2. Is the manuscript technically sound, and do the data support the conclusions?

Reviewer #2: Yes

3. Has the statistical analysis been performed appropriately and rigorously? 

Reviewer #2: Yes

4. Have the authors made all data underlying the findings in their manuscript fully available?

Reviewer #2: Yes

5. Is the manuscript presented in an intelligible fashion and written in standard English?

Reviewer #2: Yes

6. Review Comments to the Author

Reviewer #2: In my opinion, the authors dşd all the necessary corrections and now the revised version are adequate and more appropriate

7. PLOS authors have the option to publish the peer review history of their article (what does this mean?). If published, this will include your full peer review and any attached files.

Reviewer #2: **Yes: **ERTUG AYDIN

---

## [Editor Report · Acceptance letter]

13 Jul 2023

PONE-D-23-08153R1 

New failure mechanism for evaluating ultimate inclined load adjacent to slope 

Dear Dr. Xu:

I'm pleased to inform you that your manuscript has been deemed suitable for publication in PLOS ONE. Congratulations! Your manuscript is now with our production department. 

Kind regards, 

on behalf of

Dr. Ahmed Mancy Mosa 

Academic Editor

PLOS ONE